# Interaction between Chromodomain Y-like Protein and Androgen Receptor Signaling in Sertoli Cells Accounts for Spermatogenesis

**DOI:** 10.3390/cells13100851

**Published:** 2024-05-16

**Authors:** Kuo-Chung Lan, Yin-Hua Cheng, Yun-Chiao Chang, Kuo-Ting Wei, Pei-Ling Weng, Hong-Yo Kang

**Affiliations:** 1Department of Obstetrics and Gynecology, Kaohsiung Chang Gung Memorial Hospital and Chang Gung University College of Medicine, Kaohsiung 833401, Taiwan; 2Center for Menopause and Reproductive Medicine Research, Kaohsiung Chang Gung Memorial Hospital and Chang Gung University College of Medicine, Kaohsiung 833401, Taiwan; 3Department of Obstetrics and Gynecology, Jen-Ai Hospital, Taichung 412224, Taiwan; 4Graduate Institute of Clinical Medical Sciences, Chang Gung University College of Medicine, Kaohsiung 833401, Taiwan; 5Department of Biological Science, National Sun Yat-sen University, Kaohsiung 804201, Taiwan; 6Division of Endocrinology and Metabolism, Department of Internal Medicine, Kaohsiung Chang Gung Memorial Hospital and Chang Gung University College of Medicine, Kaohsiung 833401, Taiwan

**Keywords:** chromodomain Y-like protein, androgen receptor, Sertoli cells, spermatogenesis

## Abstract

Spermatogenesis is a highly regulated process dependent on androgen receptor (AR) signaling in Sertoli cells. However, the pathogenic mechanisms of spermatogenic failure, by which loss of AR impairs downstream target genes to affect Sertoli cell function, remain incompletely understood. By using microarray analysis, we identified several AR-regulated genes involved in the maturation of spermatogenesis, including chromodomain Y-like protein (CDYL) and transition proteins 1 (TNP-1), that were significantly decreased in ARKO mouse testes. AR and CDYL were found to co-localize and interact in Sertoli cells. The AR–CDYL complex bound to the promoter regions of TNP1 and modulated their transcriptional activity. CDYL acts as a co-regulator of AR transactivation, and its expression is decreased in the Sertoli cells of human testes from patients with azoospermia. The androgen receptor–chromodomain Y-like protein axis plays a crucial role in regulating a network of genes essential for spermatogenesis in Sertoli cells. Disruption of this AR–CDYL regulatory axis may contribute to spermatogenic failure. These findings provide insights into novel molecular mechanisms targeting the AR–CDYL signaling pathway, which may have implications for developing new therapeutic strategies for male infertility.

## 1. Introduction

The process of sperm production, known as spermatogenesis, depends on the interactions between Sertoli and germ cells in the seminiferous tubules. Sertoli cells, also known as “mother” or “nurse” cells, provide necessary nutrients, signaling molecules, and other substances to support germ cell development [1,2]. Sertoli cells also play a role in the regulation of gene expression and epigenetic modifications.

Azoospermia can be defined using testicular histology, which may exhibit a Sertoli cell-only phenotype, testicular maturation arrest (MA), hypospermatogenesis, or normal but obstructed spermatogenesis [3]. Azoospermia is uncommon but not rare and occurs in approximately 2% of men in the general population. Most of these men are healthy, and the cause of impaired spermatogenesis is rarely identified.

MA is characterized by germ cells that do not undergo complete spermatogenic development. Maturation arrest may be complete or incomplete [4]. The incomplete type refers to the rare sperm observed on testicular biopsy or ejaculation. However, the cause of MA is unclear. Numerous MA patients may have genetic anomalies that interfere with sperm production, a phenomenon that is not comprehensively understood. The lack of suitable animal models is one of the limitations of research on the molecular mechanisms of MA [5].

The expression and significance of the androgen receptor (AR) in spermatogenesis have been identified. AR acts as a transcription factor activated by specific molecules and controls the expression of genes critical for male puberty and fertility. In males, the production of testosterone by Leydig cells is essential for the proper maturation of sperm, and this process is primarily controlled by changes in genes mediated by the AR [6]. AR is a vital regulator of the self-renewal of spermatogonia stem cells and the progression of meiosis [7]. Studies of complete or specific testicular cell type-specific AR-null mouse models show that spermatogenesis is arrested at the meiosis stage and lowers serum testosterone levels, resulting in azoospermia and infertility [8]. Mice with mutations in the *AR* gene exhibit pathological phenotypes similar to those observed in humans [9]. In Sertoli-specific ARKO mice lacking AR in Sertoli cells, spermatogenesis cannot progress beyond the pachytene or diplotene stages. However, the deletion of the *AR* gene in germ cells does not affect spermatogenesis or male fertility [10]. This suggests that androgens indirectly regulate meiosis by acting on Sertoli cells, as germ cells do not have AR function but require androgen to advance beyond meiosis. Analysis results of these mouse models contribute to elucidating the pathological process and AR signal of human maturation arrest azoospermia.

The removal of histones from chromatin and their replacement by transition proteins (TNP) is a unique epigenetic event during spermatogenesis. Transition proteins 1 and 2 (TNP1 and TNP2) are the basic chromosomal proteins in a specific time window preceding protamine deposition in the germline of mammals [11,12]. The molecular basis underlying highly orchestrated chromatin reorganization, notably the transition from histone to protamine replacement, is largely unknown [11].

The chromodomain Y (*CDY*) gene family originates from the autosomal gene chromodomain Y-like (*CDYL)* protein, also known as *CDYL1* [13]. During evolution, the *CDYL1* gene was retroposed to the Y chromosome, resulting in the creation of *CDY* and its subsequent amplification into *CDY1* and *CDY2* [13]). Although all human CDY family genes, including *CDY1* and *CDY2,* are located on the Y chromosome and abundantly expressed in the testes, the human autosomal gene *CDYL/CDYL1* is located on chromosome 6 [13]. Notably, non-primate mammals lack the Y-linked *CDY* genes. However, the mouse *Cdyl* (CDY-like) gene on chromosome 13 is a homolog of the human autosomal gene *CDYL/CDYL1* [13]. The protein products of either the human *CDYL/CDYL1* or mouse *Cdyl* genes are highly similar [13]. This protein contains a chromatin organization modifier (chromodomain) domain and a histone acetyltransferase domain. The function of human *CDYL/CDYL1* in regulating spermatogenesis has not been comprehensively elucidated. Germline deletion of the *Cdyl* gene causes teratozoospermia and progressive infertility in male mice [14]. The mouse CDYL protein is abundantly expressed in the nuclei of steps 9–12 spermatids [15], and the human CDY protein is mainly present in the nuclei of late spermatids (mature spermatids and spermatozoa), where histone hyperacetylation occurs [16]. This implies that human CDYL plays an essential role in the late maturation of spermatogenesis [17].

However, whether CDYL protein is present in Sertoli cells and contributes to regulating the expression of genes and epigenetic modifications therein is unclear. In knockout mouse models, CDYL and AR dysfunction exhibit similar spermatogenesis deficiency in later stages. Despite some progress, the relationship between AR and CDYL expression and activity is ill-understood and should be studied further to fully elucidate their functions and mechanisms. In this study, we aimed to investigate whether CDYL and AR are critical for the maturation of spermatogenesis, particularly in Sertoli cells.

## 2. Materials and Methods

### 2.1. Animals

We used the Cre-Lox strategy to generate wild-type (WT) and androgen receptor-knockout (ARKO) mice, as previously described [18]. The WT (AR^flox/Y^) and ARKO (AR^-/Y^;ACTB::Cre) mice were established by breeding female mice with AR^floxed/floxed^ genes with male ACTB-Cre mice. These mouse strains were acquired from Prof. Chawnshang Chang and Jackson Laboratory, respectively. The detailed procedures for this process are outlined in our previous publication [18]. The mice, aged 2–3 months, were randomly placed in plastic cages containing aspen bedding, with five mice per cage. They were allowed to acclimate for at least one week before the start of experiments. The mice were kept under controlled environmental conditions, including a temperature of 22 ± 2 °C, humidity maintained at 50 ± 20%, and a 12 h light/dark cycle. They had continuous access to food and water. After 3 weeks from the birth of offspring, we conducted genotyping through PCR analysis to confirm the presence of the desired genetic modifications in the WT and ARKO mice. After a 60-day period, we collected the testes from the mice for use in subsequent experiments. All animal procedures were conducted in accordance with the guidelines and were approved by the Institutional Animal Care and Use Committee of Chang Gung Memorial Hospital (IACUC No. 2016111001).

### 2.2. Cell Culture, Knockdown by Small Interferential RNAs, Plasmid, Chromodomain Y-like Recombinant Protein 

The TM4 mouse Sertoli cell line (BCRC-60254) was procured from the Bioresource Collection and Research Center (BCRC) in Hsinchu, Taiwan. TM4 cells were maintained in alpha-minimum essential medium containing penicillin (25 U/mL, Invitrogen, Carlsbad, CA, USA), streptomycin (25 μg/mL, Invitrogen, Carlsbad, CA, USA), 5% horse Serum (Thermo Fisher Scientific, Waltham, MA, USA), and 2.5% fetal bovine serum (Invitrogen, Carlsbad, CA, USA). TM4 cells were seeded at a density of 2 × 10^4^ cells/mL in 12-well plates (5 × 10^4^ cells in 6-well plates or 1.5 × 10^6^ cells in 10 cm dishes) and transfected using the TransIT-X2 transfection reagent from Mirus Bio (Madison, WI, USA), following the manufacturer’s instructions. siRNAs targeting mouse AR and CDYL were obtained from ON-TARGET plus SMART pool siRNAs by Dharmacon (L-050296-00-05, L-049239-01-05, Horizon Discovery, Cambridge, UK), and negative control was used (D-001810-10-05, ON-TARGET plus Control pool siRNAs, Dharmacon). The siRNAs were used to knock down AR and CDYL in TM4 cells for subsequent Chromatin Immunoprecipitation (ChIP) and luciferase assays.

For overexpression, a mouse AR overexpression plasmid and control plasmid (pcDNA3) were used as previously described [18]. The mouse CDYL overexpression plasmid was obtained from GenScript (Piscataway, NJ, USA). CDYL recombinant proteins were sourced from OriGene (TP508673, Rockville, MD, USA) to investigate the mRNA expression of AR, CDYL, and TNP1. TM4 cells were transfected to overexpress AR and CDYL for luciferase assays. Following transfection, cells were incubated at 37 °C, counted, and lysates were prepared at the specified time points for further analysis.

### 2.3. Human Testicular Sperm Extraction

Then, males with the control group, obstructive azoospermia, and non-obstructive azoospermia were recruited from Kaohsiung Chang Gung Memorial Hospital (CGMH) during office visits for testicular sperm extraction (TESE) and Assisted Reproductive Technology (ART). The urological services provided counseling to patients undergoing TESE using a previously described protocol [18].

Three distinct patterns were routinely identified by pathologists by histology of the seminiferous tubules through testicular sperm extraction; one group had tubules with germ cell aplasia and only Sertoli cells (indicative of SCOS), the second group had tubules with germ cell maturation arrest, and the third group had tubules containing mature germ cells (indicative of normal spermatogenesis). The diagnosis of SCOS was established if tubules with Sertoli cells alone were detected in at least two sites per testis based on testicular biopsies [18,19]. The Institutional Review Board of Chang Gung Memorial Hospital (approval number: CGMH97-2399A3, 201601485A3) has reviewed and approved this study protocol.

### 2.4. RNA Extraction and Preparation from Mice Testes and the TM4 Cell Line

Total RNA was extracted from the testes of 60-day-old WT and ARKO mice using the TRIzol reagent (Invitrogen, Carlsbad, CA, USA) and reverse-transcribed according to the manufacturer’s recommendations. The tissue was then mechanically homogenized until it completely dissolved. The purification of testes RNA followed the Quick-RNA MiniPrep Plus protocol (ZYMO RESEARCH, Irvine, CA, USA). To determine gene expression profiles and associated pathways, we conducted a microarray assay. The RNA samples from the mouse testes were first assessed for RNA integrity using a TapeStation 4200 (Agilent, Santa Clara, CA, USA), and those with a RIN (RNA Integrity Number) value of ≥7.0 were further prepared using the WT PLUS Reagent kit (Thermo Fisher Scientific, Waltham, MA, USA). For the total RNA extraction from TM4 cells, TRIzol Reagent (Invitrogen, Carlsbad, CA, USA) was used according to the manufacturer’s instructions. To assess RNA purity, A230/260 (>1.8) and A260/280 (≥2.0) ratios were measured using a NanoDrop spectrophotometer, ensuring high-quality RNA for subsequent real-time PCR analysis.

### 2.5. Quantitative Real-Time Reverse Transcriptase–Polymerase Chain Reaction (RT–PCR)

A total of 2 µg of RNA was used for cDNA synthesis, which was carried out using M-MLV Reverse Transcriptase from Promega (Madison, WI, USA) following the manufacturer’s recommendations. For the quantification and analysis of mRNA expression, real-time reverse transcription–PCR (RT–PCR) was performed using Fast SYBR^®^ Green Master Mix from Applied Biosystems (Waltham, MA, USA) and the ABI 7500 Fast Real-Time PCR System, also from Applied Biosystems (Waltham, MA, USA). To ensure accurate measurements, the content of 18S (for mouse tissue samples) and β-actin (for TM4 cell samples) was used as a normalization reference. The following primer pairs were employed: 5′-GCTGCCTTGTTATCTAGCCTCAA-3′ and 5′-AATGACCGCCATCTGGTCAT-3′ for *AR*, 5′-ACAGCATTTCATTCGCAG-3′ and 5′-CGCTATCAGAAACTTCGTG-3′ for *CDYL*, 5′-TGTGATGCGGCAATGAGC-3′ and 5′-CGACTGGGATTTACCCACTC-3′ for *TNP1*, 5′-CGCCGCTAGAGGTGAAATTCT-3′ and 5′-CGAACCTCCGACTTTCGTTCT-3′ for *18S* (mouse), and 5′-AGGCCAACCGTGAAAAGATG-3′ and 5′-TGTGGTACGACCAGAGGCATAC-3′ for *β-actin* (TM4 cell).

### 2.6. Gene Expression Profiles Determined Using Affymetrix Microarray Assay

We conducted a microarray assay to determine the gene expression profiles and underlying pathways. The RNA products were hybridized with mouse transcriptome 1.0 microarray chips and then scanned with GeneChip Scanner 3000. These assays were conducted according to the manufacturer’s standard protocols. Raw microarray data were subjected to quality control and analyzed using Partek (Partek Genomic Suite v7.0), a commercial software kit. Partek was used to perform analysis of variance (ANOVA) and calculate the *p*-value and fold change for genes. Genes with a *p*-value of <0.05 and fold change of >1.5 were determined to be differentially expressed genes. We created a heat map based on these differentially expressed genes. In addition, to understand the underlying pathways contributing to the physiological differences between the two types of mice, we conducted pathway enrichment analyses.

### 2.7. Immunohistochemical and Immunofluorescence Staining 

Testes are fixed in 10% Neutral Buffered Formalin (NBF) and embedded in paraffin. The testes sections were deparaffinized, rehydrated from graded ethanol (100%, 95%, 70%), and heated to retrieve the antigen in citrate buffer by autoclaving. Hydrogen peroxide blocking solution (Abcam, Cambridge, UK) was added to cover the sections. The slides were incubated with primary antibodies against AR (1:100, human sample use SP107, EZTA, Sierra Madre, CA, USA; 1:500, mouse sample use PG-21, Millipore, Billerica, MA, USA) and CDYL (1:1000, ab5188, Abcam, Cambridge, UK). After the appropriate secondary IgG antibody was used, peroxidase/DAB+ reagent was used to detect the bound antibody for 5 min. The slides were scanned, and images were captured using a PANNORAMIC MIDI II (3DHistech, Ltd., Budapest, Hungary). For double immunofluorescence staining, sections were retrieved and incubated with AR (1:500, PG-21, Millipore, Billerica, MA, USA) and CDYL (1:1000, ab5188, Abcam, Cambridge, UK) primary antibodies overnight at 4 °C. Antibody binding of AR and CDYL was visualized using Alexa Fluor^®^-conjugated secondary antibodies (Invitrogen, Carlsbad, CA, USA). Finally, the sections were counterstained with a fluorescence mounting Medium with DAPI (Vector Labs, Burlingham, CA, USA). All samples were visualized using a fluorescence microscope (Nikon E800, Tokyo, Japan), and images were captured and processed using an SPOTRT3 digital camera and SPOT basic software (Diagnostic Instrument Inc., Sterling Heights, MI, USA).

### 2.8. Protein Extraction of Mice Testes and TM4 Cell Line

The testes and TM4 cells line were washed with PBS and subsequently treated with ice-cold Pierce RIPA buffer (Thermo Fisher Scientific, Waltham, MA, USA). The testes were homogenized using an electric homogenizer. The contents were then agitated in microcentrifuge tubes for 30 min at 4 °C. Afterward, the tubes were centrifuged, and the resulting supernatant was collected in a fresh tube to isolate the proteins. The protein concentration was determined by performing a Bradford assay (Bio-Rad, Hercules, CA, USA). This protein extraction process was performed for co-immunoprecipitation and Western blotting analysis.

### 2.9. Western Blot Analysis of Mice Testes and TM4 Cell Line

A total of 80 µg of protein was loaded into the wells of an 8–10% SDS-PAGE gel and subsequently transferred onto a nitrocellulose membrane (Amersham Pharmacia Biotech, Bucks, UK). Following this, the membranes were blocked with 5% skimmed milk for 1 h at room temperature. Next, the membranes were incubated overnight at 4 °C with primary antibodies against AR (1:500, PG-21, Millipore, Billerica, MA, USA) and CDYL (1:1000, ab5188, Abcam, Cambridge, UK). The protein bands were then visualized using an ECL kit (Millipore, Billerica, MA, USA). For Western blot densitometry and band quantification, image analysis software (Quantity One, Bio-Rad, Hercules, CA, USA) was utilized. GAPDH levels were employed as an internal control. This software allowed for the quantification of protein levels, including the amount of protein used, primary and secondary antibody binding, and subsequent detection. Loading control measures were employed to ensure accurate quantification.

### 2.10. Co-Immunoprecipitation

A total of 6 × 10^6^ cells/mL were washed with ice-cold PBS and then collected using lysis buffer. The insoluble material was removed through centrifugation. Approximately 500 µg of total protein was used as an input control. For immunoprecipitation, 1 mg of total protein was incubated with primary antibodies. The immunoprecipitated samples were subsequently washed four times for 5 min each with a salt-free wash buffer via centrifugation. The immunoprecipitated proteins were eluted and separated using SDS-PAGE. Immunoprecipitation was conducted using Catch and Release v2.0 (Millipore, Billerica, MA, USA) to explore protein–protein interactions.

### 2.11. Chromatin Immunoprecipitation

6 × 10^6^ cells/mL were cross-linked in situ using 1% formaldehyde from Sigma-Aldrich (Burlington, MA, USA). Following cross-linking, the cells were lysed and subjected to sonication to fragment DNA into pieces ranging from approximately 300 to 1000 base pairs in size. The ChIP assay was carried out using the ChIP Assay Kit with Magnetic Protein A/G Beads, following the manufacturer’s recommendations provided by Millipore (Billerica, MA, USA). The pre-cleared chromatin was then subjected to immunoprecipitation with antibodies specific to AR (PG-21; Millipore, Billerica, MA, USA) or CDYL (ab5188, Abcam, Cambridge, UK) overnight. DNA fragments that formed complexes with the target protein were isolated and subsequently identified through quantitative PCR. This PCR was performed to validate the presence of the *CDYL* and *TNP1* genes, with each sample undergoing triple repeats. The PCR cycles consisted of an initial step at 95 °C for 20 s, followed by 40 cycles of denaturation at 95 °C for 3 s and annealing/extension at 60 °C for 30 s. The PCR primers are: 5′-TCCATTCACAGGACGCTCAC-3′ and 5′-AGTGGGTGGCTTGACATCAG-3′ for CDYL, and 5′-CCTGACACTGTCCGTTCTCA-3′ and 5′-TGTGCATGGACCCTTTAGCC-3′ for TNP1.

### 2.12. Dual-Luciferase Assay

A reporter plasmid containing the enhancer sequence of the CDYL or TNP1 promoter was constructed through the Gene Synthesis Service provided by Genelabs Life Science (Taipei, Taiwan). Subsequently, these constructs were cloned into the pGL4-Promoter plasmid from Promega (Madison, WI, USA). For the assessment of transcriptional activity, assays were conducted using the dual-luciferase assay system (Promega, Madison, WI, USA) following the manufacturer’s guidelines. Cells at a concentration of 2 × 10^4^ cells/mL were transfected with either overexpression or siRNA genes, and they were co-transfected with the *CDYL* or *TNP1* promoter plasmids along with Renilla plasmids. Transfection was facilitated using the TransIT-X2 transfection reagent from Mirus Bio. Twenty-four hours after the plasmid transfection, the cells were lysed, and the activity of both Firefly and Renilla luciferase was measured using a luminometer (Lumat LB950, Berthold, Wildbad, Germany).

### 2.13. Statistical Analysis

GraphPad software Prism 5.0 (GraphPad Software) was used to analyze homogeneous data. For homogeneous data, unpaired, two-tailed Student t tests were used to evaluate the statistical differences between the AR knockout and wild-type mice. A parametric analysis of variance (ANOVA) with Dunnett’s test was conducted to assess the statistical differences between the *AR,* or *CDYL* siRNA-treated or overexpressed groups and the control group. *p* < 0.05 was considered statistically significant, and the data are presented as mean ± standard deviation (SD).

## 3. Results

### 3.1. Identification of High-Throughput Profiling of Androgen Receptor-Regulated Genes in Spermatogenesis Using Microarray Analysis

To determine androgen-regulated genes relevant to the control of the maturation process of spermatogenesis, global gene expression was analyzed by microarray analysis. After data preprocessing, 3591 upregulated and 4286 downregulated genes were identified in the testes of ARKO mice compared with that in control mice. The expression values were hierarchically clustered using the heat map package, and the color contrast indicated that there were significant differences between the gene expression of ARKO and control mice (Figure 1A). Microarray analysis, followed by gene ontology profiling, identified several functional groups of genes, including cellular components, biological processes, and molecular functions. These gene lists were analyzed for GO within the principal group of biological processes (Figure 1B). The top-ranked candidate genes responding to sustained deficiency of AR levels in the mouse testes are presented in Table 1, which shows that several genes related to spermatogenesis, including *CDYL*, *TNP1* gene, the outer dense fiber of sperm tails 1(*Odf1*), and spermatogenesis-associated 6 (*Spata6*) and maelstrom homolog (*Mae1*), are significantly decreased. Notably, the expression of the sex-determining region Y-box 9 (*Sox9*) gene was significantly increased, as we previously reported [18].

### 3.2. AR Is Colocalized and Interacts with CDYL in Sertoli Cells 

To further examine the expression pattern of CDYL and AR in spermatogenesis, immunohistochemical analysis was first used to investigate the localization of AR and CDYL in male ARKO and control mouse testes. All Sertoli cell nuclei, peritubular myoid cell nuclei, and Leydig cells displayed AR-positive staining in the wild-type mouse testes (Figure 2A). Sertoli cell nuclei, spermatogonia, spermatocytes, and spermatids showed CDYL distribution in the wild-type mouse testes (Figure 2A). Using immunofluorescence staining, both AR and CDYL were expressed and co-localized in the nucleus of wild-type mouse Sertoli cells (Figure 2B).

Since both AR and CDYL proteins were expressed in testicular tissues from wild-type mice and immunoprecipitated with AR or CDYL antibodies, the interaction complex was then analyzed by Western blotting. We also observed that AR interacted with CDYL in testicular tissues (Figure 2C).

Consistent with a previous report, ARKO mice exhibited developmental arrest of germ cells in the absence of AR immunostaining. Notably, less prominent immunostaining of CDYL was detected in Sertoli cells and spermatogonia of ARKO mouse testes with deficient spermatogenesis (Figure 2A). Compared with the testes from the control group, CDYL protein and mRNA expression were significantly decreased in testicular tissues of ARKO mice by Western blot and real-time PCR (Figure 2D,E).

### 3.3. AR and CDYL Regulate Downstream AR-Targeted Genes

To explore the mechanism of AR signaling on downstream targeted genes involved in spermatogenesis, we applied the ALGGEN-PROMO virtual lab to predict putative AR response elements (AREs) in the CDYL and TNP1 promoter regions (Figure 3A,B). To determine whether AR binds to the targeted gene promoter regions, TM4 cells were knocked down for AR or CDYL expression, followed by chromatin immunoprecipitation (ChIP) assays (Figure 3A,B). The results showed that insufficient expression of AR significantly decreased AR binding to ARE in the CDYL and TNP1 promoter regions, and similar data for CDYL were obtained (Figure 3A,B). Next, CDYL or TNP1 upstream regions starting at position −3000 were cloned into the pGL3-Basic plasmid before the luciferase reporter gene. The siRNA-mediated knockdown of AR or CDYL expression substantially decreased CDYL or TNP1 promoter activity in TM4 cells, as shown by dual-luciferase assays. Downregulating AR or CDYL also decreased CDYL and TNP1 promoter activity (Figure 3C). In contrast, the overexpression of AR or CDYL increased the CDYL and TNP1 promoter activity (Figure 3D).

### 3.4. Investigation of the mRNA Expression of AR, CDYL, and TNP1 Association by CDYL Rescue in Sertoli Cells

To determine whether AR and CDYL affect *TNP1* gene expression, we first transfected TM4 cells with control, AR, or CDYL siRNA and treated them with or without recombinant CDYL protein for 24 h. Subsequently, the mRNA expression of *AR*, *CDYL*, and *TNP1* was analyzed in the different treatment groups of Sertoli cells (Figure 4 and Appendix A). As shown in Figure 4A, *AR* mRNA expression was significantly downregulated by AR or CDYL siRNA-treated groups compared with that in the control group in TM4 cells; similar results were obtained for *CDYL* mRNA levels (Figure 4B). However, neither AR nor *CDYL* mRNA was significantly altered in the presence of recombinant CDYL protein. Notably, *TNP1* mRNA expression was significantly downregulated in the AR and CDYL siRNA-treated groups. However, AR and CDYL siRNA-treated groups showed significant TNP1 upregulation in the presence of recombinant CDYL protein in TM4 cells compared with that in the control group without recombinant CDYL protein treatment (Figure 4C). These results suggest that AR interacts with specific genes in spermatogenesis, notably CDYL and TNP1. Moreover, CDYL is a downstream effector of AR that effectively affects the gene regulation of *TNP1*.

### 3.5. Decreased Expression of CDYL in the Human Testes from Patients with Azoospermia

We examined the expression of AR and CDYL in normal and pathological human testes (Sertoli cell-only syndrome and maturation arrest) by immunochemistry (Figure 5). As shown in Figure 5A,D, both AR and CDYL proteins were detected in normal human testes (Figure 5A,D). While the AR protein was expressed, the CDYL protein expression was negligible in the testes of patients with Sertoli cell-only syndrome (Figure 5B,E). Similar results were obtained, showing that a low level of CDYL protein was expressed in human testes with maturation arrest (Figure 5C,F). These results demonstrate that CDYL expression is correlated with normal human spermatogenesis and is deficient in patients with maturation arrest.

## 4. Discussion

Understanding the mechanisms by which Sertoli cells respond to androgen signals and transmit them to germ cells is of utmost importance. However, the specific mechanisms by which the AR activates downstream targeted genes to affect the function of Sertoli cells remain poorly understood. The AR protein is not detected in germ cells but is expressed in the nuclei of most Sertoli, Leydig, and peritubular myoid cells in both human and mouse testes. In contrast, the CDY family of genes exhibits diverse expression patterns in human and mouse testes. All human CDY family genes are highly expressed in the testes, and their normal expression is crucial for complete spermatogenesis [20]. In this study, we observed low expression of CDYL in human Sertoli cells, early-stage spermatogonia, and spermatocytes with deficient spermatogenesis in the testes of patients with Sertoli cell-only syndrome and maturation arrest, similar to its expression in ARKO mouse testes. Notably, AR and CDYL were both highly expressed in Sertoli cells. Moreover, the combination of AR and CDYL signaling in Sertoli cells is necessary for the activation of spermatogenesis genes, and CDYL plays a crucial role in regulating the ability of AR to activate genes involved in sperm production. Additionally, a decrease in AR expression in Sertoli cells can downregulate the ability of CDYL to activate the AR-targeted genes involved in spermatogenesis.

Consistent with our findings, Gill-Sharma et al. proposed that functional deficits in testosterone reduced CDYL protein expression and that testosterone impacts a common chromatin remodeling mechanism during spermiogenesis through the chronological expression of different molecules required for this biological process [21]. The detailed epigenetic mechanisms have not been investigated further. The CDYL/AR complex activates the downstream *Cdyl* gene, triggers chromatin remodeling by histone acetylation, and is involved in spermatogenesis.

A significant change in testicular cells during the later stages of spermatogenesis is the compaction of DNA in the nucleus. Unlike typical DNA packaging, sperm DNA is not coiled or associated with nucleosomes but is tightly packed into the sperm head through the use of unique germ cell chromatin proteins. Small, basic protamines replace standard histones typically associated with DNA, and transition nuclear proteins (TNPs) play a crucial role in this process. It was previously believed that TNPs displace histones and are subsequently replaced by protamines. However, TNPs may not be necessary for histone displacement or protamine deposition. Nevertheless, chromatin condensation, DNA break repair, and protamine P2 processing were impaired in double-knockout mice [22], indicating that TNPs play essential roles in these processes. On the other hand, while germ cells migrate into the adluminal compartment, tight junctions (TJs) develop between adjacent Sertoli cells located behind the germ cells, establishing the blood–testis barrier (BTB) [23,24]. The BTB segregates meiotic and postmeiotic cells into the immunologically privileged adluminal compartment. Tight junctions (TJs) are specialized anchoring junctions composed of several integral and peripheral membrane proteins, including claudins (Cldn) 1, 2, 3, 4, 5, 7, 8, and 11, which have been demonstrated to be expressed in the testis [25]. Among them, Cldn3 is a transcriptional target of AR in Sertoli cells. The mRNA levels of Cldn3 not only decrease in mice with Sertoli cell-specific ablation of AR but also increase in the TM4 Sertoli-like cell line following androgen treatment. Moreover, protein levels of Cldn3 exhibit the highest expression levels of AR in Sertoli cells according to immunostaining assays. This suggests that Cldn3 is a direct transcriptional target of the androgen receptor, as identified through microarray gene expression profiling to detect genes with altered transcript levels in a mouse model for conditional androgen insensitivity [26]. Differences in the expression of Sertoli cell genes known to be involved in junction formation, such as Cldn 3 and 11, could be observed in Sertoli cells lacking AR (SCARKO mice) [27]. However, Masutaka Tokuda et al. indicated that CDH1 expression is only specific in undifferentiated type A spermatogonia in the mice testes, as shown by histological analysis [28]. Xiaoyu Xia et al. have also reported CDH1 to be expressed sequentially in the testis, which plays important roles in spermatogonia maintenance and spermatogonial stem cells [14].

The function and target genes of the CDYL protein in diseases are poorly understood, and the role of this protein in normal and pathological testes is not comprehensively understood. CDYL has histone acetyltransferase activity in both human and mouse proteins [15]. Notably, the germline deletion of the Cdyl gene in mice caused abnormal patterns of histone methylation and acetylation in the testes, resulting in the defects in spermatogonia maintenance and spermatozoon morphogenesis [14]. Imbalances in spermatogenic gene expression have been associated with male infertility. Moreover, recent research has shown that histone replacement is disrupted in the testes of Cdyl transgenic mice [29] and strengthens the CDYL-regulated histone modifications in spermatogenesis. This study found that AR binds to the *CDYL* gene promoter and affects TNP expression, and CDYL functions as a cofactor with AR in Sertoli cells. These findings suggest that the histone-to-protamine exchange mechanism may depend on the combined action of AR and CDYL in Sertoli cells, providing additional insight into the mechanism of spermatogenic failure.

During evolution, autosomal CDYL proteins acquire diverse protein functions by acquiring new exons into the Y chromosome [13,15]. As it is located in the AZF region of the human Y chromosome, CDY1 has a Y chromosome microdeletion related to dyszoospermia [30,31]. In human testicular tissue sections, Kleiman et al. [20] divided the tissue samples into four groups: normal spermatogenesis, hypospermatogenesis, spermatocyte maturation arrest, and Sertoli cell-only syndrome, and the results showed that the expression level of CDY2 was downregulated in the latter three groups, and was closely related to the spermatogenic disorder. Moreover, CDY1 downregulation was relatively severe, almost undetectable in the SCO group; our previous report also confirmed that CDY1a and CDY1b double deletions were observed in eight patients with SCO using Y-chromosome array-based comparative genomic hybridization [30]. These results suggest that the deletion of both CDY1 and CDY2 leads to spermatogenic failure and male infertility, and CDY2 and CDY1 functions may be redundant in the early and later stages of spermatogenesis, as a single missing copy of CDY (CDY1A deletion or CDY1B deletion) seems to have no effect on male fertility. One possible explanation is that the products of the *CDY1* and *CDY2* genes have the same origins, as high as 98% in gene homology, and their expression location and periods overlap, which may have a dose compensatory effect [32]. However, the presence of CDYL proteins may not be sufficient to provide functional compensation for *CDY* gene deletion as CDYL was detected even in MA patients (Figure 5), suggesting that it plays a distinct role in Sertoli cells other than germ cells of the testes [20]. Our results collectively suggest the novel role played by CDYL in the regulation of AR signaling in Sertoli cells.

## 5. Conclusions

In conclusion, we reported a previously unidentified relationship between AR and CDYL. Therefore, our results indicate that CDYL functions as a transcriptional coregulator for AR transactivation and plays an integral role in the androgen/AR signaling pathways involved in regulating Sertoli cell function and male fertility. Collectively, our findings potentially offer a novel direction in the search for new therapeutic strategies in patients with azoospermia.

## Figures and Tables

**Figure 1 cells-13-00851-f001:**
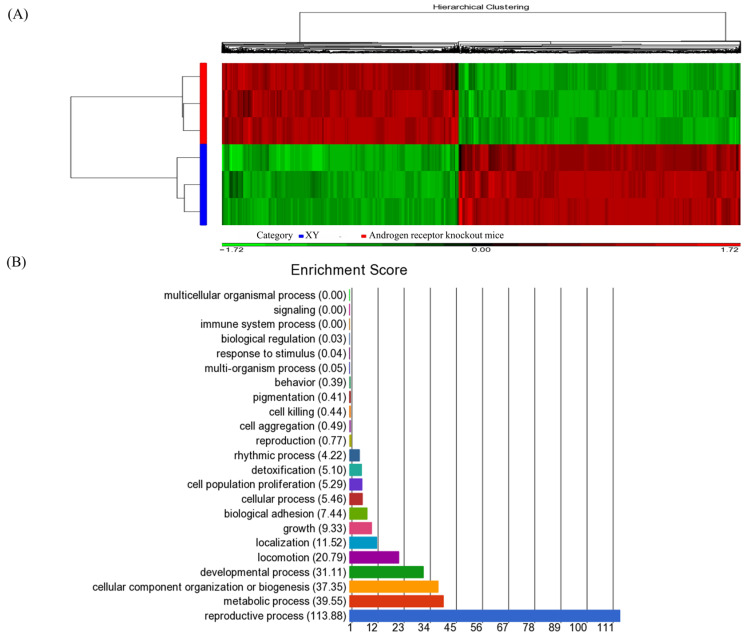
High-throughput profiling of androgen receptor-regulated genes in spermatogenesis using microarray analysis. (**A**) Red and green represent over twofold upregulated and downregulated genes, respectively. (**B**) Microarray experiments were performed in triplicate. Gene Ontology (GO) classifies gene product functions through structured and controlled vocabularies. It is available for a broad range of species. (*n* ≥ 3) in each group.

**Figure 2 cells-13-00851-f002:**
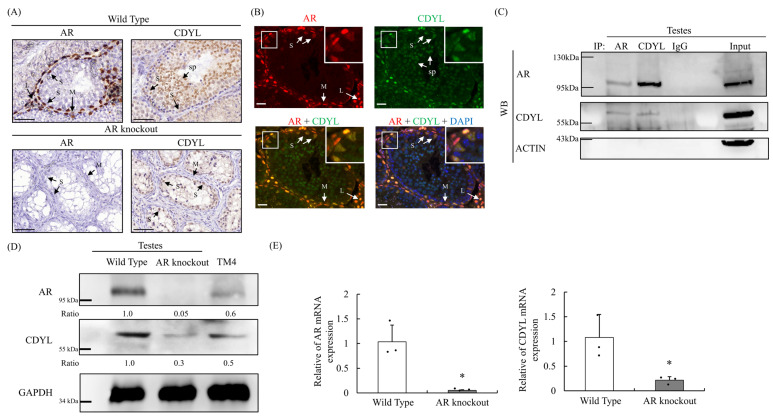
AR is colocalized and interacts with CDYL in male testes and Sertoli cells. (**A**) Immunohistochemical staining of AR and CDYL in testes obtained from wild-type and ARKO mice. Bar = 50 um. (**B**) Localization of AR and CDYL (upper) and co-localization of AR, CDYL, and DAPI (bottom) in the testes obtained from wild-type mice by immunofluorescence analysis. Bar = 20 μm. (**C**) Interaction was observed between AR and CDYL in wild-type mouse testes. IgG was used as the control for Western blotting in each group. “Input” means the sample on 10% of volume used for IP. (**D**) Protein expression patterns of AR and CDYL in wild-type mouse testes and ARKO mice by Western blotting. GAPDH was used as the internal control. (**E**) *AR* and *CDYL* mRNA expressions were detected in testicular tissues between wild-type and ARKO mice by quantitative RT–PCR assay. (*n* ≥ 3) * *p* ˂ 0.05, by unpaired two-tailed Student *t* tests was significant compared with the control. Data are expressed as the mean ± standard error of three samples per group. S: Sertoli cell; L: Leydig cell; M: myoid cell; SP: spermatids.

**Figure 3 cells-13-00851-f003:**
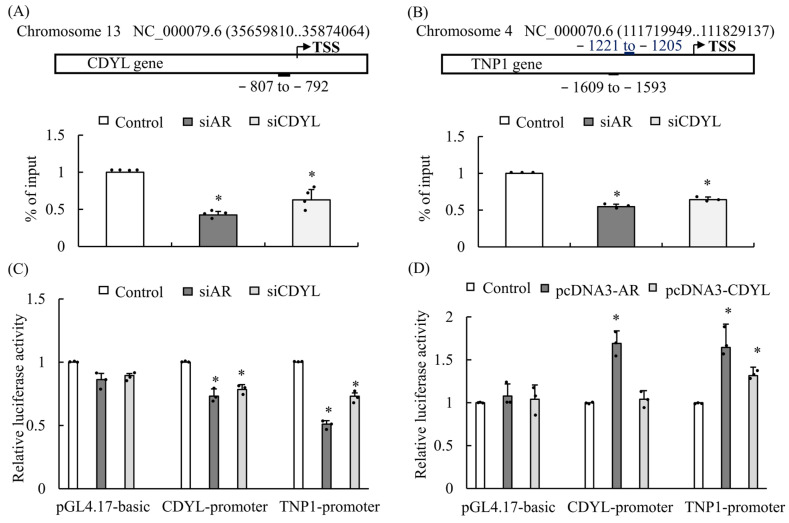
AR and CDYL regulate downstream AR-targeted genes. ChIP and luciferase assays demonstrate the *CDYL* (**A**) or *TNP1* (**B**) genes in the presence of AR binding sequences. Schematic of a putative binding site for the transcription factor, androgen-responsive elements (ARE) sequences on the *CDYL* or *TNP1* promoter. Promoter occupancy of AR on *CDYL* or *TNP1* promoter by ChIP. *CDYL* or *TNP1* upstream regions starting at position −3000 were introduced into the pGL3-Basic plasmid before the luciferase reporter gene. The siRNA-mediated knockdown (**C**) or overexpression (**D**) of AR or CDYL substantially changed the CDYL or TNP1 promoter activity in TM4 cells through dual-luciferase assays. * *p* ˂ 0.05 for one-way ANOVA was significant compared with the control. (*n* ≥ 3) in each group, and error bars represent ± SD.

**Figure 4 cells-13-00851-f004:**
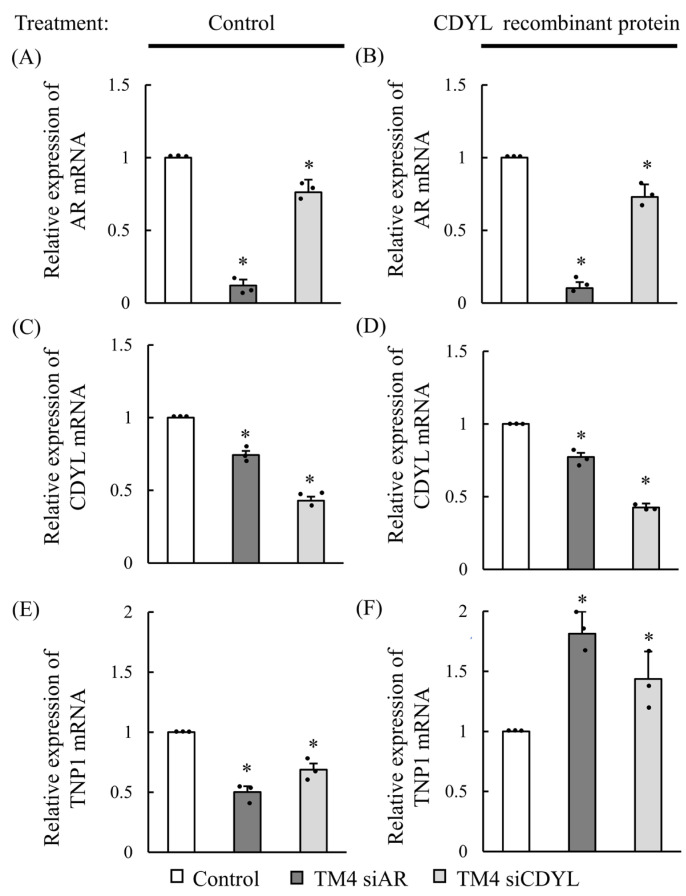
mRNA expression of *AR*, *CDYL*, and *TNP1* association by CDYL rescue in Sertoli cells. mRNA expression of *AR*, *CDYL*, and *TNP1* association among control, *AR* siRNA-treated, and *CDYL* siRNA-treated groups in TM4 cells by non-treat and CDYL rescue. mRNA expression of *AR* (**A**,**B**), *CDYL* (**C**,**D**), or *TNP1* (**E**,**F**) were detected after being transiently transfected with control, *AR*, or *CDYL* siRNA-treated (**A**,**C**,**E**). Moreover, these three groups were treated with 200 ng CDYL recombinant protein (**B**,**D**,**F**) for 24 h in TM4 cells. * *p* ˂ 0.05 for one-way ANOVA was significant compared with the control. (*n* ≥ 3) in each group, and error bars represent ± SD.

**Figure 5 cells-13-00851-f005:**
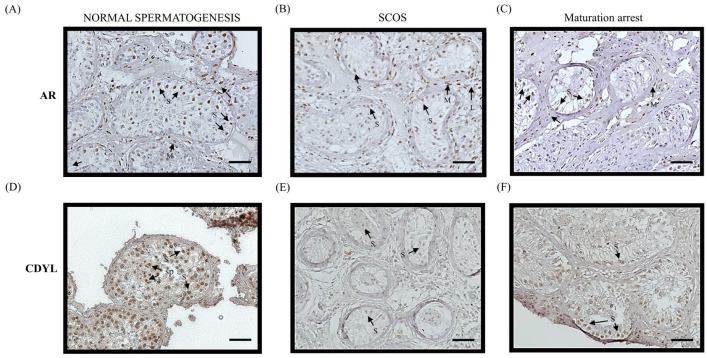
Decreased expression of CDYL in the human testes from patients with azoospermia**.** Immunohistochemistry indicated the expression of AR and CDYL protein in testicular histology from obstructive azoospermia (active spermatogenesis) and non-obstructive azoospermia (defective spermatogenesis), including Sertoli cell-only syndrome (SCOS) and maturation arrest (halted at the primary spermatocyte stage). Bar = 50 μm. The AR and CDYL signals in patients with normal group (**A**,**D**), Sertoli cell-only syndrome (**B**,**E**), and maturation arrest (**C**,**F**) (*n* = 1 patient in every group). S: Sertoli cell; L: Leydig cell; M: myoid cell; SP: spermatids.

**Table 1 cells-13-00851-t001:** Microarray gene list: spermatogenesis gene expression analysis of testis microarray.

Gene Symbol	Gene_Assignment	GO Function	RefSeq	Fold-Change
Tnp1	Tnp1 // transition protein 1	spermatogenesis	NM_009407	−291.762
Odf1	Odf1 // outer dense fiber of sperm tails 1	spermatogenesis	NM_008757	−154.729
Spata6	Spata6 // spermatogenesis associated 6	spermatogenesis	NM_026470	−5.60714
Mael	Mael // maelstrom homolog (Drosophila)	spermatogenesis	NM_175296	−4.37476
Cdyl	Cdyl // chromodomain protein, Y chromosome-like	spermatogenesis	NM_001123386	−2.70621
Sox9	Sox9 // SRY (sex determining region Y)-box 9	spermatogenesis	NM_011448	2.91174

## Data Availability

The datasets of the present study are available from the corresponding author upon request.

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
