# Peer review of "Interaction between Chromodomain Y-like Protein and Androgen Receptor Signaling in Sertoli Cells Accounts for Spermatogenesis"

_cells, 2024, doi:10.3390/cells13100851_

Round 1
Reviewer 1 Report
Comments and Suggestions for Authors
This is an interesting work points to elucidate the AR-CDYL regulatory axis and how it may contribute to spermatogenic failure.
Major Points:
- Figure 2C: Input sample should be included in the figure. It would be useful to understand how much protein you can IP with your Ab (Ab efficiency).
- Figure 2D: How can authors explain that CDYL levels in TM4 are lower than WT whole organs? In the latter should be several cell types. How can they exclude that most of the protein was from other cell types?
- Figure 2E: In many cases, relative mRNA expression does not correspond to protein levels. Why did authors not quantify WB as well? This maybe correlates with the previous point.
- Figure 4D: If author downregulate and overexpress CDYL at the same time, I would expect a kind of rescue, depending on relative efficiencies. Why does it not happen in this case?
- Row 308 paragraph: How many patients? What about the statistics here? To me, as showed in this paragraph, this experiment is meaningless.
- Row 323: Which cohort study? No information are provided.
In general, all the original blots provided should include a merge image made by HRP signal and colorimetric scan of the protein marker generated from the “machine”. Authors provided hand-made marker copied to the HRP signal…too prone to author error.
Minor Points:
- Figure 2B: A crop or zoom of the colocalizing spots (few) would be appreciated.
Reviewer 2 Report
Comments and Suggestions for Authors
Kuo-Chung Lan et al have reported that AR and CDYL interact with each other and orchestrate the process of spermatogenesis. This manuscript also showed that these genes regulated various downstream targets of spermatogenesis. Overall, the research design is rational and the results are solid. I have a few comments that need to be addressed.
1. Scale bars are missing in Figure 5.
2. All the graphs should be better to be presented in the dot plot method to show the individual value.
3. I am curious whether AR deletion in SC (Sertoli cells), has effects on BTB (blood testes barrier). It would be better to have some data on Cdh2 or Cldn1.
4. Moreover, how does CYDL affect the process of meiosis in the absence of AR in SCs?
Reviewer 3 Report
Comments and Suggestions for Authors
This research provide in sights into novel molecular mechanisms targeting the AR-CDYL signaling pathway, which may have implications for developing new therapeutic strategies for male infertility to investigate the interaction between chromodomain Y-like protein and androgen receptor signaling in Sertoli cells accounts for spermatogenesis. English writing is good, so I want to suggest this paper as "accept in present form"
Round 2
Reviewer 1 Report
Comments and Suggestions for Authors
The manuscript has been sufficiently improved, therefore, it is now suitable for publication.